# Advances in Characterizing the Transport Systems of and Resistance to EntDD14, A Leaderless Two-Peptide Bacteriocin with Potent Inhibitory Activity

**DOI:** 10.3390/ijms24021517

**Published:** 2023-01-12

**Authors:** Adrián Pérez-Ramos, Rabia Ladjouzi, Marius Mihasan, Radja Teiar, Abdellah Benachour, Djamel Drider

**Affiliations:** 1ICV-Institut Charles Viollette, UMR Transfrontalière BioEcoAgro 1158, University Lille, INRAE, University Liège, UPJV, YNCREA, University Artois, University Littoral Côte d’Opale, 59000 Lille, France; 2Biochemistry and Molecular Biology Laboratory, Faculty of Biology, Alexandru Ioan Cuza University of Iasi, Carol I Blvd, no. 20A, 700506 Iasi, Romania; 3U2RM-Stress and Virulence, UNICAEN, Esplanade de la Paix, 14000 Caen, France

**Keywords:** enterocin DD14 (EntDD14), ABC transporters, resistance to bacteriocins

## Abstract

Enterocin DD14 (EntDD14) is a two-peptide leaderless bacteriocin produced by the *Enterococcus faecalis* 14 strain previously isolated from meconium. This bacteriocin is mainly active against Gram-positive bacteria. Leaderless bacteriocins do not undergo post-translational modifications and are therefore immediately active after their synthesis. As a result, the cells that produce such bacteriocins have developed means of protection against them which often involve transport systems. In this and our previous work, we constructed different mutants deleted in the genes involved in the transport functions, thus covering all the supposed components of this transport system, using *Listeria innocua* ATCC 33090 as the indicator strain to assess the activity of externalized EntDD14. We also assessed the self-resistance of the WT and all its engineered derivative mutants against EntDD14, provided extracellularly, in order to evaluate their self-resistance. The results obtained highlight that the ABC transporter constituted by the DdG, H, I, and J proteins contributes to EntDD14 export as well as resistance to an external supply of EntDD14. Our results also have established the essential role of the DdE and DdF proteins as primary transporters dedicated to the externalization of EntDD14. Moreover, the in silico data showed that DdE and DdF appear to assemble in a formation that forms an essential channel for the exit of EntDD14. This channel DdEF may interact with the ABC transporter DdGHIJ in order to control the flow of bacteriocin across the membrane, although the nature of this interaction remains to be elucidated.

## 1. Introduction

Bacteria compete with other microorganisms in their ecological niches for nutrients and other biotic and abiotic resources, and many have developed mechanisms enabling them to outcompete their rival congeners. One of these is the production of bacteriocins, which are ribosomally synthesized antimicrobial peptides [1,2,3] produced by Gram-negative and Gram-positive bacteria, as well as by Archaea [1,4]. Bacteriocins produced by Gram-positive bacteria are currently the most studied, particularly the bacteriocins produced by lactic acid bacteria (designated as LAB-bacteriocins). Usually, LAB-bacteriocins possess narrow spectra and therefore are active against phylogenetically close bacteria. Nonetheless, some LAB-bacteriocins can target phylogenetically distant bacteria [5].

Bacteriocins are synthesized as precursor peptides, in which the core peptide is preceded by an N-terminal leader peptide. These precursor peptides are known to be inactive and must be stripped of their leader peptide to produce the active form of the bacteriocin [6]. Some of these bacteriocins can undergo enzymatic post-translational modifications (PTM), leading to post-translationally modified peptides (RiPPs) [7]. Accordingly, bacteriocins can be classified into two main classes: class I, containing RiPPs; and class II, containing unmodified bacteriocins [8]. These bacteriocins are low-molecular-weight (<10 kDa) thermostable peptides. There is also a third class of high-molecular-weight, thermolabile bacteriocins [9,10].

However, there is a relatively small number of bacteriocins that are synthesized without a leader peptide and are therefore active after their translation in the cytoplasm [11]. These leaderless bacteriocins (LLB) were discovered by Cintas and co-workers following the characterization of an enterococcal bacteriocin named L50 [12]. As can be predicted from the considerable number of genes constituting the genetic determinants of bacteriocins, their synthesis requires a mechanism involving modification, transport, and immunity proteins and in some cases additional accessory proteins [13]. Genes coding for bacteriocins are generally organized as operons, located either on plasmids or in the chromosome, and their expressions are usually co-regulated [13]. The transport of these bacteriocins is most often mediated by an ATP-binding cassette (ABC) transporter system [14] and in some cases by a secretory pathway [15]. The transport of leaderless bacteriocins is not yet fully understood. A first study conducted on aureocin 70 showed the involvement of an ABC type transporter named AurT, which is responsible of the externalization of the LLB aureocin A70 [16], while NisT is the main transporter for nisin [17]. The ABC transporter can help mature the bacteriocin by cleaving the leader peptide, or it may be carried out with the participation of an associated protein [6,14].

Bacteriocin-producing bacteria are protected against the toxicity of their own products by immunity mechanisms [18]. Most of them, as well as many bacteriocin’s mode of action, are not well understood. Nevertheless, some mechanisms have been characterized and can be mediated by specific proteins, such as that of the enterocin CRL35, where its immunity protein interacts with the forming pore, blocking it [19] or that of nisin, where a NisI protein anchored to the extracellular membrane is able to hijack nisin molecules [20]. Another mechanism involves the abovementioned ABC transporter that basically expels the surrounding bacteriocin across the membrane. This is the case of NisEFG for nisin [21], or As-48EFGH for the cyclic enterocin AS-48 [22].

EntDD14 is a leaderless bacteriocin composed of two highly related peptides [23], which is produced by *Enterococcus faecalis* 14, a strain previously isolated from meconium [24]. Recently, we have demonstrated that the EntDD14 transport system is mediated by an ABC transporter system [25] and, more importantly, by two proteins, DdE and DdF, carrying pleckstrin homology domains (PHb2) [26].

These findings related to EntDD14 transport were unexpected and suggested a mode of transport different from those described to date. To gain more insight on the EntDD14 transport systems, we compiled a collection of mutants of the genes which we assumed to have a role in the transport and assessed their activity against *Listeria innocua* as the indicator organism to evaluate the externalization of this bacteriocin by different combinations of transport systems.

In this work, we provide more information on the transport of EntDD14, through genetic evidence based on the construction of several mutant strains. We show that partial or total deletion of the ABC transport system does not completely abolish the transport and externalization of the bacteriocin to the extracellular medium. It is clear that the transport of the leaderless bacteriocin EntDD14 is strongly dependent on proteins DdE and DdF.

## 2. Results

### 2.1. The EntDD14-ABC Transporter System Contains Four Proteins

The ABC transporter associated with EntDD14 synthesis was initially described as containing at least three genes corresponding to *ddH*, *ddI*, and *ddJ* genes [25]. Deletion of the *ddI* gene significantly altered the transport of EntDD14, as the quantity of bacteriocin in the medium was reduced by ~75% [25]. To determine the role of the other components of the EntDD14-ABC transporter system, different mutant strains were constructed, namely the ΔddG, the triple ΔddHIJ, and the quadruple ΔddGHIJ mutant strains, and assessed for their respective antibacterial activity against *L. innocua* ATCC 33090, used as a sensor and indicator strain. The total antibacterial activity obtained with the WT strain was used as the reference value for 100% of the activity (Figure 1). Thus, the culture from the triple ΔddHIJ or the quadruple ΔddGHIJ mutant strains showed an antibacterial activity of ~25% of the WT, similar to that obtained previously with the ΔddI mutant [25]. The culture of the ΔddG mutant strain exhibited an antibacterial activity of ~33% (Figure 1). From this, it can be concluded that DdG, DdH, DdI, and DdJ form a functional assembly that is involved in up to 75% of EntDd14 enterocin transport.

### 2.2. The Proteins DdF and DdE Are Essential for the Transport of EntDD14

As described above, the ABC transporter system of EntDD14 in its totality is constituted by the four *ddGHIJ* genes (Figure 1) as has already been identified for the transport of other bacteriocins [22,27,28]. These results suggest that the ABC transporter system is not the unique system involved in the transport of EntDD14. Therefore, partial or total deletion of the ABC transporter noticeably reduced but did not stop the transport of EntDD14 outside of the bacterial cell (Figure 1). This observation raises the question of what effect the loss of this bacteriocin transport activity could have. If the EntDD14 bacteriocin is not completely excreted into the extracellular medium, logically it must accumulate inside the producing cells where it could have a deleterious effect. Therefore, we assessed the survival of *E. faecalis* 14 and its isogenic mutant strains after 6 and 24 h of growth in GM17 (Figure 2).

After six hours of growth, no survival differences were observed between the WT and different mutant strains (Figure 2). The amount of EntDD14 produced after six hours of growth did not induce any cell toxicity. Nevertheless, after 24 h of growth, we could observe a loss of one log magnitude in in the Δ*ddE* and Δ*ddF* mutants’ survival rate (Figure 2) due to intracellular accumulation of EntDD14, strengthening the previously reported data [26]. The Δ*ddE* and Δ*ddF* mutants do not exert any anti-*Listeria* activity, in spite of their intact ABC transport system. In these genetic backgrounds, EntDD14 appeared to be trapped in the producing cells, arguing that translocation of this bacteriocin can be performed only in the presence of DdE and DdF proteins.

All of these results support the essential role of the DdEF system in the transport of EntDD14. The DdEF system is likely a simple gradient-dependent transporter, which alone is unable to ensure the entire transport of EntDD14 in the WT strain. The ABC transporter composed of DdGHIJ is an active ATP-dependent transporter, boosting the DdEF system to fully expel the bacteriocin outside the cell against a gradient concentration. These independent systems are thought to interact synergistically in the transport of EntDD14 outside of the bacterial cell.

### 2.3. The ABC Transporter DdGHIJ Is Involved in the Resistance to Extracellular EntDD14

The different constructed mutants, as well as the WT strain, were tested against a pure solution of EntDD14 in order to determine their intrinsic resistance to this extracellularly added bacteriocin. The results obtained allowed us to define three distinct groups, as shown in Figure 3. In the first group, which involves the WT strain, Δ*ddE* and Δ*ddF* mutant strains clearly exhibited the highest MIC values against extracellular EntDD14, with values of up to 120 µg/mL. The second group, which contains Δ*ddG*, Δ*ddI*, Δ*ddHIJ*, and Δ*ddGHIJ* mutants of the ABC transporter system, is characterized by an intermediate resistance, with MIC values against extracellular EntDD14 of 80 µg/mL. Finally, the third group comprising the Δ*bac* mutant, which is deficient in the synthesis of EntDD14 and the indicator strain *L. innocua* ATCC 33090, is characterized as having the lowest MIC values of 40 µg/mL (Figure 3).

To further analyze the interaction between the ABC transporter and the resistance to extracellular EntDD14, we performed growth kinetics in the presence of EntDD14 with a final concentration of 20 or 40 µg/mL. Notably, in the absence of added EntDD14, the growth of Δ*ddGHIJ* was similar to that of the WT (Figure 4A). Remarkably, at 20 µg/mL of EntDD14, the quadruple Δ*ddGHIJ* mutant showed an extended lag phase of five hours compared to the WT (Figure 4B), whereas at 40 µg/mL of EntDD14, the mutant was unable to grow (Figure 4C). These results support the hypothesis of the involvement of the ABC transporter in the detoxification of extracellular EntDD14.

The results in Figure 3 show that the strains exhibiting the highest resistance to extracellular amounts of EntDD14 are the WT and the Δ*ddE* and Δ*ddF* mutants. These results are coherent for the WT because all its transport systems that regulate the EntDD14 accumulation levels are compatible with its development; however, this is not the case for the Δ*ddE* and Δ*ddF* mutants. Indeed, these mutants are deficient in DdE or DdF proteins, indicating that the transport system comprising both proteins plays no role in the resistance to the extracellularly localized EntDD14. On the other hand, the resistance threshold of WT, Δ*ddE*, and Δ*ddF* mutants does not differ much from those obtained in mutants related to the ABC transporter (~2-fold) or the *Listeria* control strain (~3-fold). This can be explained by an unspecific effect of the bacteriocin when applied at a high concentration, causing a detergent-like effect, which has been reported for some cationic peptides [29].

The results in Figure 4 are also consistent with the ABC transport system, which is involved in resistance against EntDD14 by expelling it from the cellular membrane. This is reflected in panel B for which cells are confronted with EntDD14 extracellular concentrations (20 µg/mL) that cause problems in the fitness of the tested strains. Indeed, the WT takes about 10 h to adapt while the Δ*ddGHIJ* mutant takes 5 h more; however, once the adaptation succeeds, the cells grow in almost the same way as in the absence of bacteriocin. In panel C, the EntDD14 extracellular concentrations (40 µg/mL) are beyond what the Δ*ddGHIJ* mutant can tolerate, thus preventing it from growing; even the WT has its lag phase extended to 13 h for its adaptation and commencement of growth.

### 2.4. Predicted Structures of DdE, DdF, and DdE/DdF Show a Potential Transmembrane Channel

Both DdF and DdE have no homologs with known structures in the PDB database. Hence, Alphafold2, with its novel machine learning approach, was used to predict their structures. The best ranking structural model of DdF generated with a good confidence score (pLDDT value 78.3; maximum is 100) has the general aspect of a transmembrane channel with six mostly hydrophobic alpha-helices connecting the two predominantly hydrophilic regions on the extracellular and intracellular sides of the membrane. Viewed from the extracellular side, the helices are arranged in an incomplete circle (Figure 5) indicating that the channel is not complete. The DdE model was predicted to have a lower confidence score (pLDDT value 58.7) and consists of only two helices connecting the two predominantly hydrophilic regions. As both DdF and DdE are both independently required for the transport of EntDD14, we have attempted to model a multimeric complex using various subunit stoichiometries: DdE/DdF; (DdE)2/DdF; (DdE)3/DdF; DdE/(DdF)2; and (DdE)2/(DdF)2. A structural model with the highest confidence score (iptm + ptm value of 0.7; maximum is 1) was obtained for the DdE/DdF complex (Figure 5). In the complex, DdE and DdF hydrophobic helices are arranged in a complete circle, which looks like a functional transmembrane channel in agreement with the previous results stating that these proteins are involved in the exit and entry of EntDD14. Moreover, this formation associating the two proteins, DdE and DdF, also explains the fact that when one of them is missing, the translocation process across the plasma membrane no longer takes place.

## 3. Discussion

The transport of bacteriocins from the cytoplasm to the extracellular environment is a complex process usually involving Gram-positive bacteria in the ATP-binding cassette (ABC) transport system, which consists of multidomain membrane proteins powered by the energy from ATP binding and hydrolysis to expel bacteriocins across the membrane hydrophobic barrier [30]. In addition to the transport of bacteriocins and peptides involved in quorum sensing, the ABC transport system may have a role in resistance to bacteriocins such as nisin and bacitracin [31].

ABC transporters form one of the largest families of membrane proteins are present in different life lineages. They are able to transport small ions as well as large proteins [32]. Notably, ABC transporters can serve for outward or inward flows [33]. The proteins of the ABC transport system consist of four domains, of which two are hydrophobic transmembrane domains (TMD), which bind and transport the substrates, and two hydrophilic cytosolic nucleotide-binding domains (NBD), which are the site of ATP binding and hydrolysis [14].

The ABC transporter system is not the only system in Gram-positive bacteria to transport bacteriocins. Indeed, a number of class II bacteriocins, such as bacteriocin 31, enterocin P, bacteriocin T8, divergicin A, and lactococcin 972, can be secreted across cell membranes through the *Sec*-dependent pathway [6].

It was reported that Gram-positive bacteria can predominantly use the ABC transporters to transport bacteriocins outside the cytoplasm, and in some cases, the Sec-dependent pathway [6]. However, it would be interesting to know if there are bacteriocins capable of using both pathways. For the moment, this information is still lacking, although it has been reported in the case of bottromycin A2, which is a RiPP containing macrocyclic amidine and unusual β-methylated amino acid residues [34].

In the case of the two-peptide leaderless EntDD14, genes encoding the ABC transporters are included in a cluster of 10 genes. Genes *ddG*, *ddH*, *ddI*, and *ddJ* are most likely encoding the components of the ABC transporters involved in the transport of EntDD14. While the role of DdG encoded by the *ddG* gene remains to be determined, the other genes code for well-known products. The gene *ddH* encodes a putative efflux Resistance-nodulation-division (RND) transporter periplasmic adaptor subunit, and *ddI* and *ddJ* genes encode an ATP-binding protein of the ATPase family and a putative macrolide export ATP-binding/permease protein (MacB), respectively. Bioinformatic analyses underpinned a high percentage of identity between DdHIJ and FGH proteins, which are part of the enterocin As-48-ABC transporters. The percentages of identity were 94%, 98%, and 98%, respectively. However, the identity between DdG and E protein, which represents its counterpart from As-48, was found to be ~46%.

In respect to EntDD14, it has become evident that the ABC transporter is not the only actor taking this role. Indeed, partial or total deletion of the ABC transporter system does not abolish the transport of EntDD14 outside of the cytoplasm, arguing that another pathway independent of the ABC system is involved. This independent pathway may imply DdE and DdF pleckstrin homology domain-containing proteins that may form potential channels, as discussed above and previously reported [26]. In related to this, recently we established that *E. faecalis* 14, deficient in proteins DdE and DdF, is unable to transport EntDD14 into the extracellular medium, as the bacteriocin was not detected in the Δ*ddE* and Δ*ddF* mutant supernatant cultures. On the contrary, EntDD14 is accumulated intracellularly, exerting a toxic effect. From these data, we can conclude that there is a potential interrelationship between the two systems: the ABC transporter DdGHIJ and the novel pathway composed of DdE and DdF proteins. Such an independent but interacting system has already been reported in other leaderless bacteriocins such as aureocin A53, in which Orf7 and Orf8, which are homologous to DdE and DdF, are involved in the transport of this bacteriocin [27]. It should be pointed out that, in the case of aureocin, the alteration of the ABC transporters decreased the transport of this bacteriocin [27]. Thus, in both systems, it is clear that the ABC transporters are unable to transport the newly formed bacteriocins in the absence of DdE and DdF in the case of EntDD14, or their counterparts Orf7 and Orf8 in the case of aureocin A53. Further studies are required to decrypt the interplay between both systems and their dependence or lack thereof on the energy from ATP hydrolysis.

In respect to the contributions of the EntDD14 transporters, either the ABC machinery, which is composed of DdG, DdH, DdI, and DdJ proteins, or the novel likely channel pathway composed of proteins DdE, DdF, or DdEF with resistance to extracellular EntDD14, we show here that ABC transporter DdGHIJ plays a role with intermediate profile, but DdEF does not. Interestingly, ABC transporters of As-48, which are named AS-48EFGH, also confer resistance to this bacteriocin, when it is added extracellularly [22].

In the present work, we strengthen the former hypothesis in which the ABC transporter DdGHIJ is not the unique molecular actor involved in the transport of EntDD14. Indeed, using the triple and quadruple mutant strains constructed here (Δ*ddHIJ* and Δ*ddGHIJ*), we established that the production of EntDD14 is not fully completed, as it is in the WT strain, in spite of the expression of genes encoding this bacteriocin. An independent system constituted of DdE and DdF proteins associated in the form of a protein complex whose organizational conformation reveals a channel that can serve as an exit route for the EntDD14 is absolutely and definitely required. On the other hand, we established that the ABC transporter involved in the translocation of EntDD14 has a role in the resistance to this bacteriocin.

The phenotypic results of the mutants obtained throughout these works support the following hypothesis. To transport EntDD14 outside the cell, the bacterium uses the DdEF channel as a primary non-ATP-dependent transport up to a certain concentration of EntDD14 in the extracellular medium. After that, the bacterium will resort to the use of the ABC transporter composed of DdGHIJ which has to pump more bacteriocin into the extracellular medium. This ABC system requires ATP to transport the bacteriocin in the opposite direction of the concentration gradient. This ABC transporter is also involved in the protection of the bacterium against the bacteriocin accumulated outside the cell. This hypothesis would explain the fact that a *ddGHIJ* quadruple mutant continues to produce ~25% of the bacteriocin compared to the WT and the fact that DdE and/or DdF mutants are unable to externalize EntDD14. The two systems, DdEF and DdGHIJ, should operate synergistically to control the flow of bacteriocin across the membrane, through a mechanism that remains to be understood.

## 4. Materials and Methods

### 4.1. Bacterial Strains and Growth Conditions

All bacteria used in this work are listed in Table 1. These microorganisms were routinely grown as following: *Enterococcus faecalis* strains in M17 medium supplemented with 0.5% glucose (GM17), at 37 °C. *Escherichia coli* strains in Luria–Bertani (LB) broth at 37 °C by shaking at 160 rpm. The *Listeria innocua* ATCC 33090 strain in Brain Heart Infusion (BHI) broth at 37 °C. When bacteria carried the pLT06 plasmid or its derivatives, the medium was supplemented with chloramphenicol (Cm) at 10 µg/mL for *E. coli* or 15 µg/mL for *E. faecalis*.

### 4.2. Construction of the E. feacalis 14 Mutant Strains

The *ddG*, *ddHIJ*, and *ddGHIJ* genes were deleted from the *E. faecalis* 14 chromosomes by independent recombination events using the pLT06 plasmid [35]. The oligonucleotides utilized in this work are listed in Table 2. The full detailed protocol to achieve the construction of the mutants is provided in Pérez-Ramos et al. [26]. Briefly, the flanking regions of each gene or gene cluster were cloned into the pLT06 plasmid after the subsequent steps of PCR amplification, restriction enzyme digestion, and ligation of the PCR and plasmid fragments. The constructions were performed in *E. coli* XL1-Blue strain and then, the generated recombinant plasmids were transferred by electroporation to the *E. faecalis* 14 WT strain or Δ*bac* mutant strain as required. Afterward, the first recombinant event (the plasmid integration into the chromosome) in the cells was induced by shifting the growth temperature from 30 °C to 42 °C in the presence of Cm at 15 µg/mL. The positive colonies were subjected to the second recombinant event (plasmid excision from the chromosome) by culturing them in the absence of Cm at 30 °C. The suitable mutants for each gene or gene cluster deletions were verified by sequencing the surrounding genetic environment.

### 4.3. Antimicrobial Activity against the Indicator Strain L. innocua ATCC 33090

Overnight cultures from the WT and mutant strains, as well as purified EntDD14, were screened for their anti-*L. innocua* activity by using the spot-on-lawn s method. Briefly, a uniform layer of *Listeria* culture was deposited on a soft BHI agar (1%) plate using a swab. Then, we spot 4 µL of each culture of 2.10^9^ CFU/mL on BHI agar previously inoculated with the target strain. Then, the plates were incubated at 4 °C for 1 h and then overnight at 37 °C. The absence or presence of inhibitory zones around the spots was recorded.

### 4.4. Purification of Bacteriocin

EntDD14 was purified from the supernatant of the WT. The purification procedure was adapted from Abriouel et al. [36] and was performed as described in Pérez-Ramos et al. [26]. Briefly, a 24 h cell-free supernatant and the CM Sephadex^®^ C-25 resin (GE Healthcare Life Sciences, Chicago, IL, USA) were incubated together for 24 h at room temperature by shaking at 90 rpm. Then, using a chromatography column, the resin was settled and washed, first with distilled water and then with 0.5 M NaCl. The EntDD14 linked to the resin was eluted with 1.5 M NaCl. Afterwards, using PD MidiTrap G-10 columns (GE), the solution of EntDD14 was desalted with MilliQ water. The pure EntDD14 was then quantified with the BCA assay protein kit (Sigma-Aldrich, St. Louis, MO, USA) and dried out with the miVac Sample Concentrator (SP Scientific, Warminster, PA, USA) for storage. When used, an aliquot of pure EntDD14 was resuspended in the appropriate volume of MilliQ water to obtain the desired concentration.

### 4.5. Sensitivity of the WT and Engineered Variants to Extracellular EntDD14

The WT and all engineered variants were tested against their own bacteriocin provided extracellularly to evaluate their self-resistance. An exponential culture of each strain diluted to 10^6^ CFU/mL was spread onto a M17 agar plate using a swab. 10 µL of a pure EntDD14 solution was deposited on the plate at different concentrations: 240, 200, 160, 120, 100, 80, 60, 40, and 20 µg/mL. The plates were incubated overnight at 37 °C and then the inhibition halos were measured. In addition, the bacterial growth of the *E. faecalis* 14 WT and its Δ*ddGHIJ* mutant in a GM17 broth with and without the addition of EntDD14 at 20 or 40 µg/mL was monitored in a 96-well microplate using a SpectraMax i3 spectrophotometer (Molecular Devices, San Jose, CA, USA). The cultures were inoculated with overnight pre-cultures to achieve the same initial OD_600nm_ of 0.2. Measurements were taken every 15 min at OD_600nm_ for 19 h.

### 4.6. Alphafold2 Structure Prediction of DdE, DdF and DdE/DdF Complex

Models of DdE and DdF monomers were generated with AlphaFold2 [37] running locally via Docker (https://www.docker.com/, accessed on 15 March 2022) on an Ubuntu 20.04 workstation with the corresponding FASTA sequences (WP_086325714 and WP_086325715, respectively) as inputs and the full_dbs preset. Following AMBER99SB [38] relaxation and OpenMM [39] energy minimization, five models were generated, and the highest ranked conformation was selected based on its pLDDT confidence score. Assembly of a DdE/DdF multimeric complex at various stoichiometries was explored using AlphaFold2-Multimer [40], and the Amber relaxed models were ranked based on their weighed combination of pTM and ipTM (iptm + ptm) confidence scores. Structures were visualized in UCSF ChimeraX [41] and transmembrane topology was predicted using DeepTMHMM [42].

## Figures and Tables

**Figure 1 ijms-24-01517-f001:**
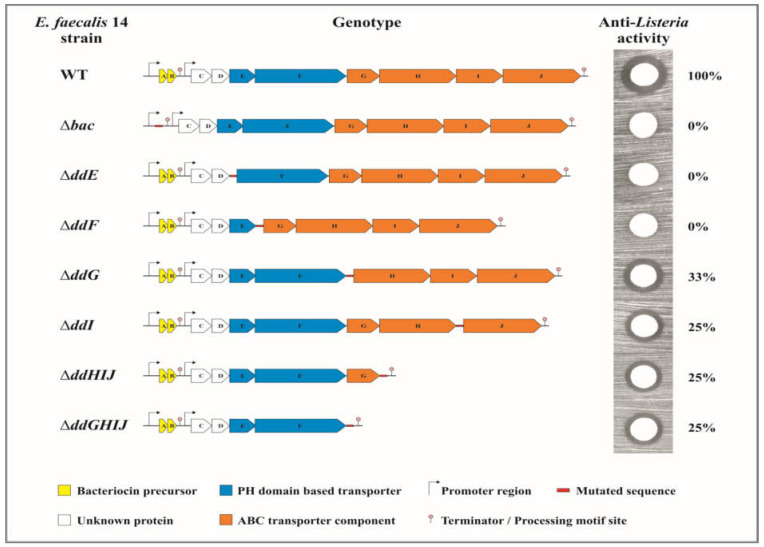
Schematic representation of *Enterococcus faecalis* 14 mutants showing their genotypes and their antibacterial activities against *Listeria innocua* ATCC 33090. A volume of 4 µL in an overnight culture ~2 × 10^9^ CFU/mL was spotted on the Brain Heart Infusion plate. The data are the means of at least three independent experiments.

**Figure 2 ijms-24-01517-f002:**
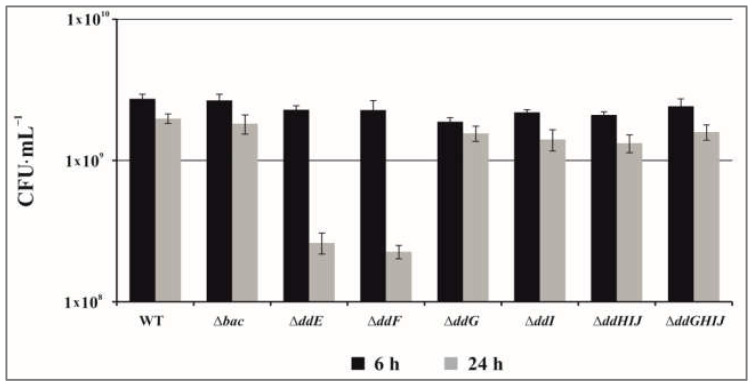
Survival of *Enterococcus faecalis* 14 and its mutant strains after 6 h and 24 h of growth in GM17. The data are the means of at least three independent experiments.

**Figure 3 ijms-24-01517-f003:**
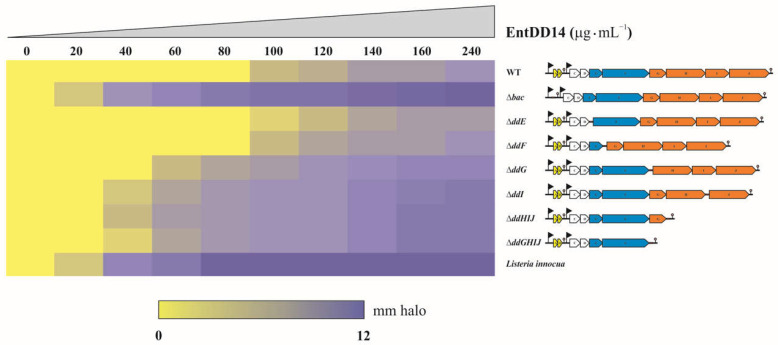
Heatmap profile of *Enterococcus faecalis* 14 and its mutant strains in GM17 broth in the presence of extracellular EntDD14. The data are the means of at least three independent experiments.

**Figure 4 ijms-24-01517-f004:**
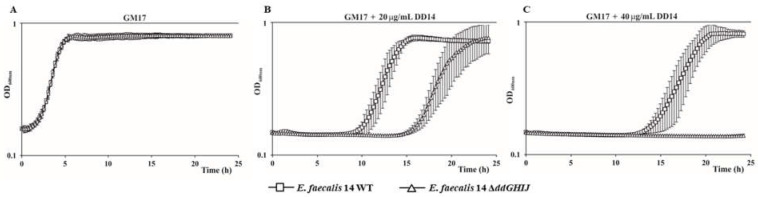
Growth curves of *Enterococcus faecalis* 14 and its Δ*ddGHIJ* mutant strain in GM17 broth (**A**) and in the presence of EntDD14 at 20 µg/mL (**B**) and 40 µg/mL (**C**).

**Figure 5 ijms-24-01517-f005:**
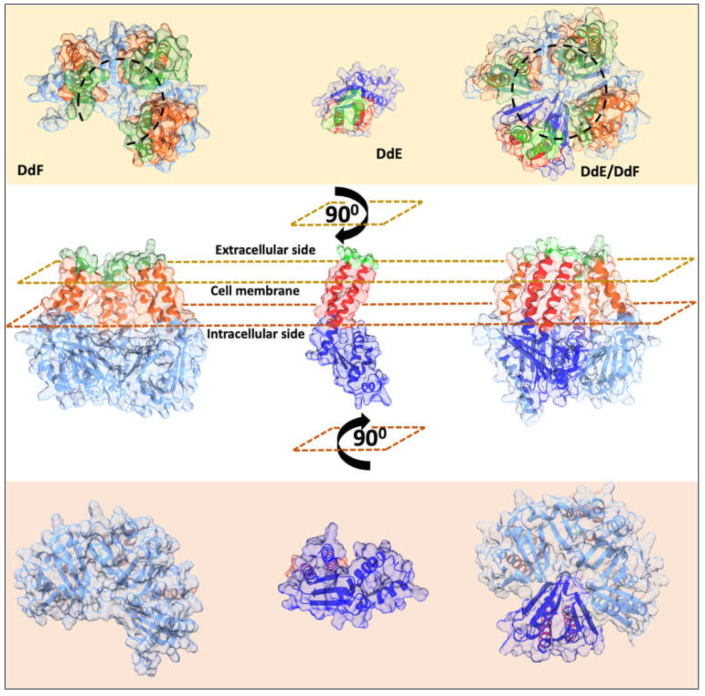
Predicted structures of DdF and DdE and their possible assembly into the DdE/DdF complex, a potential complete transmembrane channel. Transmembrane topology as predicted using DeepTMHMM is highlighted in shades of blue for intracellular regions, shades of red for transmembrane regions, and shades of green for extracellular regions. Models were visualized with UCSF Chimera.

**Table 1 ijms-24-01517-t001:** List of the bacteria used in this work.

Bacteria	Plasmids	Resistance	Characteristics	Reference
*Escherichia coli*				
XL1-Blue	-	-	Plasmid-free type strain used for plasmid cloning	Agilent Technologies
XL1-Blue [plT06]	pLT06	Cm^R^	Source of the conditioned replicative pLT06 plasmid used for mutant strategies	[25]
XL1-Blue [pLT06:Δ*ddG*]	pLT06:Δ*ddG*	Cm^R^	Derivative of pLT06 by cloning of a 2158 pb DNA fragment harboring flanked regions of *ddG* gene	This study
XL1-Blue [pLT06:Δ*ddHIJ*]	pLT06:Δ*ddHIJ*	Cm^R^	Derivative of pLT06 by cloning of a 2052 pb DNA fragment harboring flanked regions of *ddHIJ* genes	This study
XL1-Blue [pLT06:Δ*ddGHIJ*]	pLT06:Δ*ddGHIJ*	Cm^R^	Derivative of pLT06 by cloning of a 2134 pb DNA fragment harboring flanked regions of *ddGHIJ* genes	This study
*Enterococcus faecalis*				
14	-	-	Natural strain isolated from meconium	[24]
14 Δ*bac*	-	-	Deletion mutant strain of *ddAB* genes	[25]
14 Δ*ddE*	-	-	Deletion mutant strain of *ddE* gene	[26]
14 Δ*ddF*	-	-	Deletion mutant strain of *ddF* gene	[26]
14 Δ*ddG*	-	-	Deletion mutant strain of *ddG* gene	This study
14 Δ*ddI*	-	-	Deletion mutant strain of *ddI* gene	[25]
14 Δ*ddHIJ*	-	-	Deletion mutant strain of *ddHIJ* genes	This study
14 Δ*ddGHIJ*	-	-	Deletion mutant strain of *ddGHIJ* genes	This study
*Listeria innocua*				
ATCC 33090	-	-		[25]

**Table 2 ijms-24-01517-t002:** List of oligonucleotides used in this work.

Oligonucleotide	Sequence 3′-5′	Utilization	Amplicon Size (pb)
ddG 1F-PstI	ATTAAACTGCAGTTGAATTCACTCAATCATTTT	Amplification of *ddG* upstream fragment	1.133
ddG 2R-Stop	CTATCACTAGGATCCTTAGACTTACTACGATACGTCTGTTTGTA		
ddG 3F-Stop	TAAGTCTAAGGATCCTAGTGATAGGATATAGGAGAAGATAATGAGT	Amplification of *ddG* downstream fragment	1.049
ddG 4R-NcoI	ATTAAACCATGGGAACACTGATTTGGACAT		
ddG 5F	AGGAAAATGTTGATTTGGTGTTT	Outer primer; verification of the plasmid integration	-
ddG 6R	CTAGAGATTGGGTTTGTTCTTCC		
ddHIJ 1F-PstI	ATTAAACTGCAGAAATATGCTTTTTCCTTACA	Amplification of *ddHIJ* upstream fragment	1.051
ddHIJ 2R-Stop	CTATCACTAGGATCCTTAGACTTACTCATTATCTTCTCCTATATCT		
ddHIJ 3F-Stop	TAAGTCTAAGGATCCTAGTGATAGGTAAAGGCCAAAGAATTAGA	Amplification of *ddHIJ* downstream fragment	1.025
ddHIJ 4R-NcoI	ATTAAACCATGGAATTTTATCCCAAAGAAAGT		
ddHIJ 5F	ATCAGAATGTTTTCATGCGTT	Outer primer; verification of the plasmid integration	-
ddHIJ 6R	AAGTTAATGGTGATACTTCACAA		
oriF	CAATAATCGCATCCGATTGCA	Cloning verification in pLT06 plasmid	-
Ks05R	CCTATTATACCATATTTTGGAC		

## Data Availability

The *E. faecalis* 14 chromosome sequence is deposited on the NCBI database (accession number CP021161.1), as well as the DdE and DdF protein sequences (accession numbers WP_086325714 and WP_086325715, respectively).

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
