# Peer review of "Advances in Characterizing the Transport Systems of and Resistance to EntDD14, A Leaderless Two-Peptide Bacteriocin with Potent Inhibitory Activity"

_ijms, 2023, doi:10.3390/ijms24021517_

Round 1

Reviewer 1 Report

General comments:

The paper from Pérez-Ramos et al. dealt with the characterization of transporter systems for the export and immunity against the bacteriocin Enterocin DD14 (EntDD14) of Enterococcus faecalis 14. EntDD14 is a leaderless, unmodified class IIb two-peptide bacteriocin with activity against other Gram-positive bacteria, i.e. Listeria spp. and Clostridia perfringes strains. The producer strain was isolated from meconium and further characterized in previous studies. In the present work, different deletion strains of E. faecalis 14 were used to elucidate the role of the transporter genes ddEFGHIJ, which are supposedly involved in export as well as immunity for EntDD14. The authors analyzed the secretion of the two-peptide bacteriocin as well as the EntD14 susceptibility by different methods and claim that both ddEF and ddGHIJ are involved in transport and immunity. Upon deletion of the transporter DdE/DdF they claim that cell viability decreased due to intracellular accumulation of the active peptides. A hypothetical structure of the DdE/DdF channel was proposed using AlphaFold modelling.   

The paper opens interesting questions on the little investigated mechanisms of immunity and export of two-peptide bacteriocins. However, the study appears preliminary and there are several concerns that are detailed below.

1) Overall, the conclusions concerning the dual functions of the transport systems DdEF and DdGHIJ appear overstated based on the presented results. Regarding the activity levels of the mutant strains (Figure 1) and the resistance towards extracellularly added EntDD14 (Figure 3) it rather appears that DdEF is the exporter of EntDD14 while DdGHIJ acts as complex system (containing transmembrane elements and an ABC-transporter) conferring immunity. Without DdEF, no extracellular activity was detected in supernatants of the respective mutant, indicating that it is the main exporter of EntDD14. Since DdGHIJ is still present in this strain, one can exclude a major role of this system in export. The authors state that this system constitutes to 75% of the EntDD14 export, but no such activity was observed in the ΔDdEF strain. The authors state that the reduced activity in the supernatants of the ddG/ddI/ddHIJ/ddGHIJ mutants supports their claim of an active role of the system in export but important controls are missing. Since production levels often correlate to biomass and expression of biosynthesis genes, it would be needed to present growth kinetics and transcript levels of the peptide genes ddAB. Furthermore, all ddG/ddI/ddHIJ/ddGHIJ mutants show increased sensitivity towards EntDD14 (Figure 3), which might also lead to reduced production titers as shown for other bacterionc, e.g. nisin.

2) The authors claim that the deletion of ddE/F leads to reduced export of the bacteriocin and in consequence to increased intracellular accumulation. Although the data in Figure 1 and Figure 2 generally support this hypothesis, the authors did not experimentally show this accumulation, e.g. by disrupting the cells and test for intracellular activity. The data also contradicts the above mentioned hypothesis that DdGHIJ is involved in export since no reduced viability was observed for the respective mutant strains. Furthermore, a strain with deleted ddE/F and ddAB genes (ΔddE Δbac) would be a suitable control for intracellular accumulation. The authors furthermore stated, that production of the bacteriocin takes place during stationary phase, which would be rather unusual for bacteriocins and was also not experimentally shown by the authors.

3) The MIC assays in figure 3 support the above described roles of DdEF and DdGHIJ. Deletion of ddEF does not alter the sensitivity of the strains towards EntDD14 and thus rather indicate that these two proteins have a function in export of the peptide and not in protection of the cells. In contrast, deletion of ddGHIJ renders the cells more sensitive, indicating that the system and its constituents function as immunity system. These results are in line with previous results shown by the group (Ladjouzi et al), where deletion of the EntDD14 peptide genes (i.e. Δbac) leads to downregulation of ddCDEFGHIJ after 24h, rendering the cells more sensitive towards EntDD14. This effect might be due to auto-regulation by the peptides and needs to be addressed in more detail by the authors.

Specific comments:

l.108: the authors state that 75% of the EntDD14 export is achieved by DdGHIJ but no such activity was detected in the ΔddE/F strains. Please comment how this supports the hypothesis that DdGHIJ is also involved in export and not just in immunity.

l.110: please indicate the number of replicates performed and at what time-point the supernatants for the activity tests were harvested. The OD600 of the cultures is needed to correlate production to biomass.

Figure 2: please indicate the number of replicates used for the standard deviation. If possible, include strains Δbac ΔddE/F to exclude an effect by the deletion. A strain ΔddEFGHIJ could be included to have a more pronounced effect on CFU/mL.

l.134 – 135: No data was shown or cited for stationary phase production of EntDD14. Either remove or substantiate with data.

l.141 – 142: No evidence presented that EntDD14 is trapped inside the cells. Cell disruption and testing for intracellular activity might be suitable to do so.

l.143 – 146: This statement is highly speculative. The absence of an effect in the ΔddGHIJ cells cannot be used as an argument for an intracellular threshold of bacteriocin accumulation. The data only supports a mild toxic effect for the deletion of ddE/F. Plasmid-based overproduction of EntDD14 could increase an assumed accumulation in ΔddGHIJ cells.

Figure 3, l.164 - 165: please indicate the number of replicates.

Figure 4, l.178 – 179: please indicate the number of replicates.

l. 183: intracellular activity of EntDD14 was not shown by the authors. Furthermore, the whole paragraph from l.182 – 186 is based on an assumption of an intracellular accumulation threshold, which was not substantiated experimentally by the authors.

l.193 – 197: the conclusion drawn in this paragraph appears illogical. The ΔddE/F strain still possesses the DdGHIJ system and is thus as resistant as the WT, which also has the system. Figure 3 clearly shows that the DdGHIJ confers immunity against extracellular EntDD14.

l.198 - 199: the statement is not substantiated by data that the ABC system DdGHIJ prevents accumulation of EntDD14. Figure 4 shows that absence DdGHIJ renders the cells more sensitive to extracellular EntDD14.

l.237 - 241: please revise sentence, it is far too long.

l.244 – 250: the relevance of this paragraph for the discussion is unclear. Might be shortened.

l.255 – 256: Authors did not show Sec-dependent secretion of bacteriocins.

Author Response

Dear reviewer,
please find attached the answers to the comments you have rightly raised.
Sincerely

Reviewer 2 Report

Authors show how proteins DdE and DdF are involved in the transport of bacteriocin enterocin DD14, a leaderless bacteriocin, outside the cell. In order to do so, authors have generated different mutants of E. faecalis 14 knocking out different genes from the EntDD14 operon. Authors show that when ddE or ddF genes are knocked out, there is no externalization of EntDD14. Moreover there seems to be an accumulation of the bacteriocin inside the cells thus causing a toxic effect to the host. Authors also show that when ddE or ddF genes are knocked out there is a similar tolerance to external EntDD14 as with the WT strain. Authors suggest that the mode of action of EntDD14 must me somewhere inside the cell and that Dde and DdF are involved in the externalization of the bacteriocin but also in its internalization. Last they generate a potential structure of DdE and DdF together using AlphaFold that suggest that the hydrophobic helices of both proteins are arranged in a complete circle, which looks like a functional transmembrane channel, that could explain the role of the DdE/DdF complex in the exit and entry of EntDD14.

In general I think that the novelty of this work is not very significant, as most of the experiments and conclusions reached are similar or exactly the same to those from their previous work “Evidence for the Involvement of Pleckstrin Homology Domain-Containing Proteins in the Transport of Enterocin DD14 (EntDD14); a Leaderless Two-Peptide Bacteriocin”. In that work they also knocked out ddE or ddF genes and observed the lack of secretion of EntDD14 when any of the 2 proteins were not produced. They also describe a toxic effect in the absence of these 2 proteins, and there is also an attempt to predict the secondary structure of DdE and DdF. Even in the abstract the previous work is mentioned and they don´t clarify what has been done new in this one. Therefore authors first should explain: what is the novelty of this work compared to their previous one?.

Another problem that I find with this work, is that is too speculative and doesn´t show enough evidence to justify the hypothesis.

One of the hypothesis of the work is that DdE and DdF not only are involved in the secretion of EntDD14 outside the cell but also as a gate of entrance to the same bacteriocin. From an evolutionary point of view I don´t see how this would make any sense. I think if there are other examples in the literature of dedicated pumps or transporters that secrete bacteriocins or other toxic compounds outside but also pump them inside, authors should mention them. I also would like to read a justification or at least a hypothesis of what would be the meaning of this dual activity.

Another hypothesis is that the action of EntDD14 is inside the cell and, most likely, not outside. I don´t think the results generated here are enough to come to this conclusion. So far the mode of action of all Gram + bacteriocins has been described outside the cell, forming pores at the membrane. If EntDD14 has a different mode of action and is entering inside, this has to be further probed. Is it possible to treat Δbac and or Listeria with EntDD14 and then try to detect the presence of the bacteriocin (MALDI or protein gel) in the soluble fraction?.

The same applies to the accumulation of EntDD14 inside the cell in the abscense of DdE or DdF. Can this be further demonstrated. Half a logarithmic unit reduction of viable cells doesn´t seem like a lot if you take into consideration that all the bacteriocin is being accumulated inside.

I am also surprised on the high sensitivity of the WT strain to it´s own bacteriocin (MIC is just 5 fold higher compared to Listeria). What is the explanation?

Leaderless bacteriocins are known to have an unspecific mode of action (as detergents) when used at high concentrations. Has this been taken into consideration when running the activity tests?

I am quite surprised about the high sensitivity of Δbac to external EntDD14, comparable to that from Listeria. What is the explanation for this?

On top of all these scientific concerns, I also find some problems in the structure of some of the sections.

Starting with the Title, I think is quite confusing.

The introduction is a little bit disorganized and sometimes introduces sentences that, in my opinion, don´t make much sense. For example, when the action of divergicin M35 is mentioned (line 40-41). What is the point? That are bacteriocins that are unspecific?.

Leader sequence (line 43) and signal peptide (line 44) are different things.

Lines 42 to 66 is a mixture of sentences paragraphs with no order. It starts with the concept of leader sequence, jumps to the classification of bacteriocins, then goes back to leaderless bacteriocins and then to the gen organization of bacteriocins. This should be reorganized so it makes a little bit of more sense.

English should be revised as well.

Author Response

(The authors gave the same response as above.)

Round 2

Reviewer 2 Report

Looks good to me! Thanks for the responses

Author Response

Dear reviewer, 

on behalf of all the authors, we would like to thank you very much. Your expertise has helped us to improve our manuscript and the presentation of our study. 

Sincerely,